# Morphological Characterstics of the Sensilla in a Monophagous Insect: *Agasicles hygrophila* Selman and Vogt (Coleoptera: Chrysomelidae, Halticinae)

**DOI:** 10.3390/insects14060501

**Published:** 2023-05-30

**Authors:** Qianhui Chen, Shuang Li, Yingying Wang, Dong Jia, Yuanxin Wang, Ruiyan Ma

**Affiliations:** 1College of Plant Protection, Shanxi Agricultural University, Taigu, Jinzhong 030801, China; chenqianhui@163.com (Q.C.); wangyy926415@163.com (Y.W.); biodong@hotmail.com (D.J.); 2Chongqing Academy of Agricultural Science, Jiulongpo District, Chongqing 401329, China; sclishuang61@163.com

**Keywords:** *Agasicles hygrophila*, monophagous insect, sensilla, scanning electron microscope, morphology

## Abstract

**Simple Summary:**

To further explore the host recognition mechanism of the monophagous insect *Agasicles hygrophila* Selman and Vogt (Coleoptera: Chrysomelidae), scanning electron microscopy images were used to examine and compare the external morphology and distribution of the sensilla of this species in both male and female adults. The results showed that there were 12 types and 46 subtypes of sensilla, some of which displayed sexual dimorphism. We found a new type of sensor, reported here for the first time, and named it sensilla petal-shaped (Sps) based on its morphological characteristics. This sensor may be related to host recognition. In addition, the potential functions of the structure are discussed. This information extends the study of the sensorium of monophagous insects and lays a morphological foundation for the exploration of the host recognition mechanism of *A. hygrophila*.

**Abstract:**

*Agasicles hygrophila* Selman and Vogt (Coleoptera: Chrysomelidae) is the key natural enemy of *Alternanthera philoxeroides* (Mart.) Griseb, an invasive weed worldwide. To understand the morphology of *A. hygrophila* and further explore the specific host localization mechanism, scanning electron microscopy was used to observe and study the morphological characteristics of sensilla on the head appendages, tarsi, and external genital segments of *A. hygrophila*. Twelve types and forty-six subtypes of sensilla were observed. These contain various types of head appendices, including sensilla chaetica, sensilla trichodea, sensilla basiconca, sensilla coeloconica, sensilla styloconica, Böhm bristles, sensilla campaniform, sensilla terminal, sensilla dome, sensilla digit-like, sensilla aperture, and many subtypes. A new type of sensor was reported for the first time, which may be related to host plant recognition. This sensor was located on the distal segment of the maxillary palps of *A. hygrophila* and was named as sensilla petal-shaped based on its morphological characteristics. Sensilla chaetica, sensilla trichodea, and sensilla basiconca are also found on the tarsi and external genital segments. In addition, sensilla basiconica 4, sensilla coeloconica 1 and 2, sensilla styloconica 2, Böhm bristles 2, and sensilla campaniform 1 were only found in females. On the contrary, sensilla styloconica 3, sensilla coeloconica 3, and sensilla dome were only found in males. Numbers and sizes of the sensilla were also different between males and females. The potential functions related to structure were discussed in comparison with previous investigations on beetles and other monophagous insects. Our results provide a microscopic morphological basis for further research on the localization and recognition mechanism of *A. hygrophila* and its obligate host.

## 1. Introduction

*Agasicles hygrophila* Selman and Vogt (Coleoptera: Chrysomelidae, Halticinae) is a highly specialized monophagous beetle that is an important natural enemy of the worldwide invasive weed *Alternanthera philoxeroides* (Mart.) Griseb. At present, this beetle has been used for almost 50 years, and this system is one of the most successful weed biocontrol programs worldwide [1,2]. Years of host range evaluation and post-release monitoring have shown that *A. hygrophila* shows high safety in relation to most tested crops and native species, producing only slight adult feeding on some non-target species, such as *Beta vulgaris* L. and *Beta vulgaris* var. cicla, on which it does not complete its life cycle [3,4,5]. In China, *A. hygrophila* can complete most of its life cycle (except for pupation) on the related species *Alternanthera sessilis* (L.) DC [5]. The use of an alternative host poses some potential risk in terms of a reduction in the control efficiency of *A. hygrophila* on *A. philoxeroides* or on potential harm to local plants. Therefore, understanding the mechanisms of host recognition and acceptance in *A. hygrophila* is a current research need.

Host recognition behaviors of phytophagous insects include searching, landing, and contact evaluation [6]. In this process, insects mainly rely on sensory organs such as photoreceptors, chemoreceptors, and mechanoreceptors to constantly collect information [7]. Visual and olfactory sensors play a major role in detecting host cues from a distance, while the evaluation of potential host plants from a short distance relies mainly on chemical and mechanical sensors [8]. Many kinds of sensilla are widely distributed on the insect body, and they participate in this selection process [9], especially receptors in the head appendages, tarsi, and various segments of the external genital. There are several types of insect sensilla, and their microstructure and morphology are closely related to their functions. For example, the antennal sensilla of the butterfly *Heliconius erato* phyllis (Lepidoptera: Nymphalidae) sense host volatiles [10]. The beetle *Leptinotarsa decemlineata* (Say) (Coleoptera: Chrysomelidae) is intensely stimulated by contact with its host plant’s sap, based on gustatory organs on their galea and antennae [11]. The gustatory organs on antennae and maxillary palps of the moth *Manduca sexta* (Linnaeus) (Lepidoptera: Sphingidae) coordinate feeding preferences [12]. The female of *Lobesia reliquana* (Hübner) (Lepidoptera: Tortricidae) chooses its oviposition host by sensing physical and chemical stimuli of potential host plants using the terminal receptors of the tarsi [13]. Sensilla located on the external genital segments play a role in sensing and selecting suitable oviposition sites on the host and in recognition of the opposite sex during courtship [14,15,16,17,18].

Research on insect host-finding has gradually expanded from study of an insect’s external behavior to the exploration of the morphological structure and molecular recognition mechanism of the relevant sensilla. Our laboratory has made some progress on the process used by *A. hygrophila* for remote localization of potential hosts. By comparing the differences between the volatiles of host plants and non-host plants, it has been determined that volatile odors are the key factors affecting the host-searching and location behaviors of *A. hygrophila* [19]. Additionally, binding and receptor proteins involved in the recognition of volatiles were also found to be located in the tarsi and certain other body parts [20,21]. Therefore, understanding the sensory organs of *A. hygrophila*, including their microscopic morphology and structures, is needed. However, the morphological description and function of *A. hygrophila* sensilla have not yet been reported, though they may take part in the host recognition process and play a specific function.

In this study, we examined the sensilla on the head appendages, tarsi, and external genital segments of both males and females of *A. hygrophila* adults and described them here for the first time based on scanning electron microscopy (SEM) images. We characterized the sensilla’s structure, morphology, distribution, number, and size. Our results detail the structure and function of the sensilla of *A. hygrophila* and create a morphological foundation for further investigation of the mechanism behind monophagy in *A. hygrophila*, as well as increase our understanding on the relationship between insect behaviors and sensors.

## 2. Materials and Methods

### 2.1. Insect Collection

*Agasicles hygrophila* specimens were obtained from the South China Agricultural University, Guangzhou, China, and a rearing colony was established in the Biosafety and Biocontrol Laboratory of the College of Plant Protection, Shanxi Agricultural University, Taigu, Shanxi, China. *Agasicles hygrophila* were reared on fresh *A. philoxeroides* leaves in the incubator, where was maintained at 25 °C and 80% RH with a 12 h dark–12 h light photoperiod. *Alternanthera philoxeroides* was initially collected from Yuhuan County, Zhejiang, China, then grown in the glasshouse of Shanxi Agricultural University, Taigu, Shanxi, China, under standard conditions at 25 ± 1 °C, 85 ± 5% RH, and a 10 h dark–14 h light photoperiod. Newly emerged adult beetles from this colony were used in this study.

### 2.2. Scanning Electron Microscopy (SEM) Procedures

Newly emerged *A. hygrophila* adults (six males and six females) were selected randomly from the laboratory colony, and the antennae, labial palps, maxillary palps, tarsi, and external genital segments of males and females were removed using tweezers and scalpels under a continuous variable time stereomicroscope (M165C, Leica, Germany). Excised parts were immediately soaked in 30% alcohol (Sinaopharm Group Chemical Reagent Co. Ltd., Shanghai, China), and then, using the ultrasonic cleaning machine (KQ118, Kunshan, China), the antennae, labial palps, and maxillary palps samples were cleaned for 15–30 s. The tarsus and external genital segment samples were cleaned for 100–120 s. Each body part was then dehydrated in a graded alcohol series (30, 50, 70, 80, 90, 100%, repeated three times). Specimens were allowed to dry under room conditions for 15–18 min. Dried samples were then mounted on holders with conductive adhesive and sputtered with gold using a small ion sputtering instrument (SBC-12, KYKY TECHNOLOGY Co. Ltd., Beijing, China) for 30–60 s. Lastly, the prepared samples were examined using a scanning electron microscope (Hitachi S-4800, HITACHI High Technologies, Tokyo, Japan) with the acceleration voltage set at 1–3.0 kV. Sample preparation and observations were performed in the electron microscope room of the Institute of Microelectronics, Chinese Academy of Sciences.

### 2.3. Statistical Analysis

The naming of sensilla types was based on the classification system of Schneider [22], Zacharuk [23], Dyer and Seabrook [24], and Crook et al. [25]. The morphological terminology and classification of the sensilla used here followed Tarumingkeng [26], Zacharuk [9], and Shanbhag and Mfller [27] in regards to the type and subtypes of sensilla based on sensilla morphology, size, and distribution. The length, base diameter, quantity, and distribution of each type of sensillum from *A. hygrophila* adult specimens were determined using Nano Measurer (Department of Chemistry, Fudan University, Shanghai, China). Each type of sensillum was measured on 30 individuals, and all data were based on the frontal views of structures to avoid any influence from shooting angles. Images were processed with Photoshop CS6 (Adobe Systems Software Ireland Ltd., San Jose, CA, USA). The number of each type of sensillum in the head appendages and tarsi was counted, and the density of each sensillum on the external genital segments was calculated per unit area. Data were analyzed with Student’s *t* tests in SPSS 26.0 (International Business Machines Corporation, IBM, Armonk, NY, USA) to determine whether differences were significant (*p* < 0.05).

## 3. Results

### 3.1. Typology, Characteristics, and Distribution of Sensilla in the Head Appendages

The antennae of *A. hygrophila* were typical moniliform structures with 11 segments: the scape, pedicel, and nine segmented flagellomeres. The flagella gradually decreased in length in the apical direction, and the last segment was nearly round. The pedicel was the shortest part, and the scape was long and swollen. The epidermises of the pedicel and scape were both scaly. There were basal fossae at the joints of each segment, indicating that each segment of the antenna was telescopic (Figure 1A). The length of total antennae of female *A. hygrophila* was significantly longer than that of the male (*t* =1.53, *df* = 58, *p* < 0.05). Seven types of sensilla were present on the antennae in both sexes, composed of twenty five subtypes, including sensilla chaetica (four subtypes: Sch.1,2,3,4), sensilla trichodea (six subtypes: St.1,2,3,4,5,6), sensilla basiconica (five subtypes: Sb.1,2,3,4,5), sensilla coeloconica (three subtypes: Sco.1,2,3), sensilla styloconica (six subtypes: Sty.1,2,3,4,5,6), Böhm bristles (two subtypes: BB.1,2), and sensilla campaniform (Sca.1). There were significant differences in the types of antennal sensilla between males and females: sensilla basiconica (Sb.4), sensilla coeloconica (Sco.1,2), sensilla styloconica (Sty.2), and sensilla campaniform (Sca.1) were only found in females, while sensilla styloconica (Sty.3) and sensilla coeloconica (Sco.3) were only found in males (Table 1). Additionally, there were more types of antennal sensilla in females than males: twenty-five subtypes were found in females and twenty-two subtypes in males.

The labial and maxillary palps were part of the mouthparts on the head (Figure 2A), and the surface was cataphracted (Figure 2B,C). The labial palps had three segments. The first and the second segments were hammer-shaped, and the third segment was tapered with a blunt end (Figure 2B). In terms of both genders, the length of segments in females was shorter than in males (*t* = −12.11, *df* = 58, *p* < 0.05) (Table 2). The maxillary palps had four segments, located on the palpifer of the lateral to stem segment. The surface of segments 2–4 had pores, and these pores had different diameters. The basal segment was short and columnar. The second segment was the longest, the base of segments 2–3 was smaller than the terminal, and the end segment was conical and blunt (Figure 2C). The palps of females were longer than those of males (*t* =27.02, *df* =58, *p* < 0.05) (Table 3).

Ten types of sensilla were observed on *A. hygrophila* labial and maxillary palps. These types of sensilla were composed of twenty-three subtypes, including sensilla chaetica (four subtypes: Sch.1,2,3,4), sensilla trichodea (three subtypes: St.1,2,3), sensilla basiconca (seven subtypes: Sb.6,7,8,9,10,11,12), Böhm bristles (BB.3), sensilla campaniform (two subtypes: Sca.2,3), sensilla terminal (two subtypes: S.te.1,2), sensilla dome (Dom), sensilla digit-like (Sdi), sensilla aperture (Sa), pores, and a newly discovered sensor tentatively named sensilla petal-shaped (Sps) based on its morphological characteristics (Figure 2). In male and female adults, the types of sensilla in the labial and maxillary palps were the same, the number of sensilla were lower than other parts, and fewer types of sensilla were different in quantity (Table 2 and Table 3).

#### 3.1.1. Sensilla Chaetica (Sch)

Sensillum chaeticum (Sch) were straight, spiny in appearance, and had a sharp tip. There were irregular longitudinal ridges from base to tip. Sch on the antennae were located in the open epidermal depression. According to the differences in morphological details, Sch were divided into five subtypes.

Sch.1 were observed as ridges on the surface that were without holes and slightly curved from the base to the tip, and the apical was blunt, which was different from other subtypes. Sch.1 were distributed at the end of each segment of the antennae flagella, always located on the top of the protrusion epidermis, and arranged in a circular pattern at the distal edge of flagellum segments 2–8 (Figure 1B). Sch.1 were also presented on the second segment of labial palps and the third segment of the maxillary palps (Figure 2K,T).

Sch.2 possessed longitudinal stripes on the surface with a sharp tip (Figure 1B). Sch.2 were distributed throughout each segment of the whole antennae, growing slanted along the direction of the antennae apical, almost close to the cuticle. This type of sensillum was located on the second segment of the labial palps and the third and fourth segments of the maxillary palps (Figure 2L,T).

Sch.3 were observed to have a smooth surface, were flat and close to the epidermal wall, and had a sharp tip. This type of sensillum was mainly found in the scape and the pedicel of the antennae (Figure 1D).

Sch.4 were distinguished by longitudinal ridges on the surface and an erect base, and the apical was slender, sharp, and curved toward the surface at nearly 90°. Sch.4 were found in the middle and lower parts of antennal flagellum segments 8–9 (Figure 1C) and in the maxillary palps (Figure 2M).

Sch.5 were observed to have a shallow ridge near the base. The base socket was compact, gradually tapering into the tip, and the tip was upright without holes. Sch.5 grew nearly perpendicular to the epidermis and were located on the inside tip of the second segment of the labial palps (Figure 2K).

#### 3.1.2. Sensilla Trichodea (St)

Sensillum trichodeum (St) were hairy, tapered or perforated at the tip, and nearly straight or slightly curved. A small number of subtypes dilated near the tip as an arrow or were dully rounded at the tip. St found in the head appendages can be divided into seven subtypes according to the morphological details.

St.1 were observed to have blunt round tips with a slightly concave apical hole. The base had a relatively open socket, and its tip bent away from the antenna surface. The longitudinal ridge on the wall spread from the base to the tip of the tapering apical. St.1 were presented on each segment of the antennae (Figure 1B) and segments 2–4 of the maxillary palps (Figure 2I,T).

St.2 were shorter than St.1, with a rounded, blunt tip which had an apical hole. Its base was inserted into an open and deep socket. There were deep longitudinal ridges on the wall, and the overall growth direction was close to the epidermal wall. This type of sensillum was mainly presented on the antennal flagella (Figure 1E).

St.3 were observed to have a round or oval apical hole, with a petal-like apical edge. The base was inserted into a compact socket, and the ridges from the apical to the basal cuticle gradually became shallow, which were mainly presented on the antennal flagella (Figure 1F).

St.4 were observed to have a round, blunt tip, and there was an open basal socket at the base. The surface was smooth, the basal area significantly coarser than the tip, and it was bent towards the surface of the antenna, with the tip parallel to the epidermis. St.4 were mainly presented on the antennal flagella (Figure 1B).

St.5 were observed to have a round, blunt tip, almost parallel to the epidermis. Walls of this type of sensillum have ridges, but these were not obvious on the tip, with concave single holes, and the other parts were morphologically similar to St.4. St.5 were found on the 8–9 segments of the antennal flagellum (Figure 1B,C) and on the labial and maxillary palps (Figure 2M).

St.6 were found to have a curved or upright tip toward the epidermis. The single top hole was larger than other Sts. This sensillum’s surface was smooth and formed a tapered tip. Its base was inserted into an open socket. St.6 were mainly presented on the antennal flagella (Figure 1C).

St.7 were distinguished by single holes on the tip, which curved toward the epidermis. The base was inserted into a loose socket parallel to the epidermis. The wall had shallow ridges, tapering from the base to the tip. St.7 were found on the labial and maxillary palps (Figure 2K,T).

#### 3.1.3. Sensilla Basiconica (Sb)

Sensillum basiconicum (Sb) was thick, with some similar to nails. The tip was blunt or thin, not hollow, and shorter than St. The base was slightly wider or had a round table-shaped uplift of the base. In the center of the base, there were various forms of small cones with an obvious concave cavity, and the surrounding film slightly sank. The wall was thin, and the outer wall was densely porous. The forward inclination angle was different among the subtypes. According to the differences in morphological details, Sb were divided into twelve subtypes.

Sb.1 were upright and grew parallel with the surface, the diameter gradually increased from apical to basal, the surface was covered with wall pores, and the density of the holes from apical to basal gradually decreased. The tip was pointed without holes, the base was slightly raised, and there was a circle of shallow marks at the junction of the cone and base. Sb1 were presented on the antennal terminal segment flagella (Figure 1G).

Sb.2 were slightly curved with different sizes of terminal holes, and the edge of the holes was petal-like. There was a large round base, of which the diameter was slightly less than the length of Sb.2. Other shapes were the same as Sb.1. This type of sensillum was located on the antennal terminal segment (Figure 1G).

Sb.3 were curved, with dense wall holes. The density of the holes gradually decreased from the tip to the base. Sb.3 had slightly blunt tips. Its base has a round platform and the base socket was slightly raised. The epidermis was located at the junction of the cone, and the base was wrinkled and had a circle of shallow marks. Sb.3 were found on the antennal terminal segment (Figure 1I).

Sb.4 were distinguished by a top single hole, with a gap that was high on both sides. Other features were similar with Sb.2. Sb.4 were found on the antennal terminal segment (Figure 1J).

Sb.5 sensilla had a shallow socket at the apex with an irregular apical margin and grew parallel to the surface of the antenna; other morphologies were similar to Sb.2. Sb.5 sensilla were found on the antennal terminal segment (Figure 1K).

Sb.6, compared with other Sbs, had a more wrinkled epidermis, large sockets in the base, and holes in the tip. Sb.6 sensilla were mainly found on the labial palps and the distal segment of the maxillary palps (Figure 2D).

Sb.7 also had a wrinkled epidermis with a depression from the base to the tip, and the base had a deeper socket than Sb.6. Sb.7 sensilla were mainly located on the upper part of the lower side of the labial palps and on the maxillary palps of adult females (Figure 2D).

Sb.8 were distinguished by a smooth epidermis, a single hole on the tip, and a base without a socket or round base; they were mainly located on the upper lateral part of the labial palps and on the maxillary palps of adult males (Figure 2E).

Sb.9 had small spikes on the epidermis, a smooth wall, blunt tip, and a base diameter that was much larger than the tip. Some were close to the epidermis and some had their tips pointing away from the epidermis, and they were distributed on the maxillary palps (Figure 2F).

Sb.10 had a smooth wall with a circle of large, shallow sockets at the base. The tip was twisted and without a hole, and the epidermis at the base was wrinkled. Sb.10 were distributed on the maxillary palps (Figure 2G).

Sb.11 had a pointed tip with no holes. The epidermis was wrinkled, and the base’s characteristics were the same as Sb.10. Sb.11 were located on the distal part of maxillary palps (Figure 2D).

Sb.12 had a smooth wall, a loosely rounded base, and curved ends growing close to the epidermis, which tapered from base to tip. This type was only located on the second segment of the maxillary palps (Figure 2H).

#### 3.1.4. Sensilla Coeloconica (Sco)

Sensilla coeloconica (Sco) were located in a shallow circular cavity formed by an epidermal depression, with a central sensory cone. This sensory cone had a pointed tip and thick base, resembling a chrysanthemum, and was surrounded by cilia with pointed tips. Based on differences in morphological details, the Sco group was divided into three subtypes.

Sco.1 were distinguished by multiple distinct grooves in the upper half, with a nonporous surface. The grooves continued to the tip and gradually became sharp. The shape was finger-like. The tip was jagged, with six protruding ridges. The base was smooth and there was a shallow socket at the junction between the base and the round platform. Sco.1 grew parallel to the surface or slightly bent. Sco.1 were only found in the female antennal flagella (Figure 1S).

Sco.2 were upright, and the smooth region near the base was shorter than that of Sco.1. The tip was smoother, and there were more protruding ridges (nine) than in the other subtypes. The base of the socket was deeper than others and the surface was smooth. Other features were similar to Sco.1. Sco.2 were only present on the female antennal flagella (Figure 1T).

Sco.3 emerged from the concave region of the cuticle, were slightly curved, and grew almost parallel to the surface of the antenna. The irregular concave region at the base was shallower and smaller than in Sco.2. The thickness of the whole sensor was not uniform, but other features were the same as Sco.1. Sco.3 were only present in the male antennal flagella (Figure 1U).

#### 3.1.5. Sensilla Styloconica (Sty)

Sensilla styloconica (Sty) were generally disk-like in form, with an upward, protruding epidermis with sensory cones; the base was thicker than the tip, which was blunt or thin, and the whole structure was either closed or open. Sty were mainly found on the shallow annular socket formed after the uplift of the antennal epidermis. Based on morphological details, six Sty subtypes were recognized.

Sty.1 were cone-shaped nails that emerged from the raised cuticle of the antenna. The walls of Sty.1 sensilla had five ridges and the sensor base appeared to have two layers. There was a pore at the blunt tip, and the tip was similar to that of Sco.1. The Sty.1 sensilla were found on the terminal segment of the antennae of both males and females (Figure 1M).

Sty.2 were spinous, short, straight, tapered structures with perforated tips. The base of the Sty.2 sensilla were mounted in loose, round structures that emerged from the antennal surface, which itself was smooth. Sty.2 sensilla were located mainly on the first segment of the female antenna (Figure 1N).

Sty.3 sensilla had a basal cone smaller in diameter than the base of Sty.2, and the socket of Sty.3 was deeper than that of Sty.2. The other features of Sty.3 were similar to those of Sty.2. Sty.3 sensilla were found only on the seventh segment of the male antennal flagella (Figure 1O).

Sty.4 sensilla had a round hole in the center of the tip, and the edge of the round hole was irregular and petal-shaped; both of the sensilla arose from a raised loose socket. The base of the socket was as thick as the apical end. Sty.4 sensilla were slightly curved with a smooth surface and were mainly found on the lower sides of the ninth segment of the female flagellum (Figure 1P).

Sty.5 sensilla had round holes at the edge of the tip, and there were about four small cylindrical protrusions over the remaining surface of the hole; Sty.5 sensilla had loose sockets at the base, which tapered gradually from a basal to a smooth apical surface; Sty.5 sensilla were found mainly on the antennal terminal segment of both males and females (Figure 1Q).

Sty.6 sensilla had a shallow round hole in the center of the tip, which had irregular petal-like shapes around the hole. The base of Sty.6 sensilla was a loose round platform tapering from base to tip and had a smooth surface. This sensillum grew perpendicular to the antenna and was found on the antennal terminal segment of both males and females (Figure 1R).

#### 3.1.6. Böhm Bristles (BB)

Böhm bristles (BB) were located mainly on the scape of the antennae and on the maxillary palps. Their shape was upright and smooth without holes. The tip was pointed or smooth. Some sensilla had a basal socket with a wider articular fossa. Based on length and the tip shape of these sensilla, they were divided into three subtypes.

BB.1 were found clustered in two spots on the upper and lower faces of the scape, with a blunt round tip similar to a ball. The base fit in a loose socket; either the tip and base were thick with a thin middle or the whole sensillum was thick and cylindrical. These sensilla had a smooth surface, and the exocuticle of the sensilla was also smooth; these sensilla stood almost perpendicular to the antennal surface. BB.1 sensilla were located on the joint between the head and the scape, as well as on the intersegmental membrane between the scape and pedicel (Figure 1V).

BB.2 were distinguished by one or three sulci at the tip; one sulcus was scoop-like and embedded. The base was thicker than the tip, while other features were the same as BB.1 (Figure 1W,X).

BB. 3 had protrusions on its epidermal tip and a loose basal socket. Other parts of this type of sensillum had a smooth epidermis. This sensillum tapered from the base to tip and was located on the medial portion of the first segment of the maxillary palps, with only one observed (Figure 2N).

#### 3.1.7. Sensilla Campaniformia (Sca)

These campaniform sensilla (Sca) were located on a small round concavity of the epidermis; they varied in form in terms of their shape, being loose, mastoid, smooth, or spherical, and either slightly protruding from the epidermis or being parallel to the epidermis. The terminals of their nerve cells extended to the epidermis, and some nerve cell terminals extended beyond the epidermis. Based on morphological details, Sca sensilla were divided into three subtypes.

Sca.1 sensilla were located in a concave region of the surface of the first segment of the female antennae, and they were smooth and had gaps along their edges. They were oriented almost parallel to the surface of the antennae, and were only found on the first segment of the female antenna (Figure 1L).

Sca.2 sensilla penetrated into the epidermis; these sensilla had no margin gap, but their other morphological characteristics were the same as Sca.1. Sca.2 sensilla were located outside the base of the first section of the labial palps, next to Sdi sensilla, and on the female maxillary palps (Figure 2S,T).

Sca.3 sensilla were distinguished by a compact basal socket. The center of the sensilla was a spherical depression, but other morphological characteristics were the same as Sca.1; they were found only on the end of the second segment of the male labial palps (Figure 2L).

#### 3.1.8. Sensilla Terminal (S.te)

Based on morphological details, S.te sensilla were divided into three subtypes.

S.te.1 sensilla had neither a socket nor round platform at their base. Their epidermis was nonporous and had a series of longitudinal ridges; the sensillum’s tip had either a hole or no hole. This sensillum was mainly located on the end of the labial palps (Figure 2P).

S.te.2 sensilla were chrysanthemum-like with a smooth epidermis. The tip’s hole was blunt and of different sizes. Other features were the same as S.te.1. The S.te.2 sensilla were located on the end of the maxillary palps (Figure 2D,Q).

#### 3.1.9. Sensilla Dome (Dom)

Sensilla dome (Dom) were shaped as a sphere in a shallow socket with a round, blunt tip with a hole in the middle. The epidermis of the sensilla was smooth, and Dom sensilla were only found in the upper part of the first section of the labial palps of males (Figure 2R).

#### 3.1.10. Sensilla Digit-like (Sdi)

Digit-like (Sdi) sensilla had a smooth surface and a plate-like wall. The tip was pointed with an apical hole; only the end was free, with all other parts attached to the surface. Sdi sensilla were found in the lateral part of the third section of the labial palps and the lower, lateral part of the fourth section of the maxillary palps (Figure 2S,T).

#### 3.1.11. Sensilla Aperture (Sa)

Sensilla aperture (Sa) formed a narrow band within a pitted area. The tip was nonporous and the surface was smooth. Sa were located on the first segment of the maxillary palps (Figure 2T).

#### 3.1.12. Sensilla Petal-Shaped (Sps)

Sensilla petal-shaped (Sps) had a grooved epidermis, were oriented parallel to the surface, and were snowflake-like in shape. Sps sensilla were found on the upper and lower sides of the first segment of the maxillary palps (Figure 2D,O).

### 3.2. Typology, Characteristics, and Distribution of Sensilla on the Tarsus

The morphologies of the protarsus, mesotarsus, and posterior tarsus of *A. hygrophila* adults were consistent, belonging to Cryptopentamera. The fourth tarsite was hidden between the third and fifth tarsites. The third tarsite was enlarged in the shape of a two-valve spoon, and the apical was wider than the base (Figure 3). There were holes on the surface of the tarsus. There were 14 types of tarsal attachment pads on the ventral surface of the first to third tarsi. The shapes of the tarsal attachment pads of males and females were significantly different, with the attachment pads of males being spoon-shaped and those of females hook-shaped (Figure A1).

The sensilla of all three tarsi were of the same type and were concentrated on the dorsum and sides of the tarsi. There were three types of sensilla of four subtypes: sensilla chaetica (two subtypes: Sch.1,2), sensilla trichodea (St.1), and sensilla basiconica (Sb.9) (Figure 3). Although the number and size of tarsal sensilla in males and females were different (Table A1 and Table A2), their location and types were almost the same.

### 3.3. Typology, Characteristics, and Distribution of Sensilla in the External Genital Segments

The venter of the external genital segments of *A. hygrophila* was wider and the apex was narrower. The external genital segments of both males and females were the eighth and ninth segments of the abdomen. The external genital segments of female *A. hygrophila* were used for laying eggs; the ninth segment is usually extended during egg laying (Figure 4A). The male’s external genitalia were generally not exposed; its aedeagus was typically contracted inside the ninth segment. The eighth segment of the male had a sulcus on the ventral surface (Figure 4B), which was the most significant morphological characteristic of males.

Both males and females had sensilla on segments eight and nine on the dorsal side of the abdomen. There were three types of sensilla on the external genital segments for both sexes, with one additional sensillum subtype on the male (St.8), which gradually bends toward the epidermis from the base to the tip. The surface of the St.8 sensillum was smooth. From the base to the end, it gradually became thinner towards the tip, with no basal socket or round platform (Figure 4F). The other sensilla on both males and females were sensilla chaetica (Sch.1), sensilla basiconca (Sb.9), and sensilla styloconica (St.1, St.5, St.6). The numbers and distribution of sensilla in males and females were different (Table A3).

## 4. Discussion

In this study, the external morphology and distribution of the sensilla of *A. hygrophila* adults were examined for the first time. In addition to 11 previously known types of sensilla, a new sensillum was identified on the maxillary palp and named as sensilla petal-shaped (Sps) due to its appearance. These sensilla were found on the antennae, mouthparts, tarsi, and external genital segments of *A. hygrophila*. Their types and numbers varied greatly according to the different body parts and beetle sexes.

### 4.1. Types and Functions of Sensilla

Sensilla basiconca (Sb), sensilla trichodea (St), and sensilla chaetica (Sch) were found on the antennae, mouthparts, tarsi, and external surfaces of the genital segments of *A. hygrophila* in both adult males and females. Among these sensilla, Sb had the most subtypes (12 in total). Sb sensilla have porous surfaces and grooves, are typical of thin-walled olfactory sensilla, and are considered to perceive olfactory cues from volatiles from the insect’s host plant [28]. For example, Sb sensilla on the antennal flagellum of the cicada *Scaphoideus titanus* Ball (Hemiptera: Cicadae) can detect olfactory stimuli [29]. Sb sensilla on insect antennae may also play a role in sensing intraspecific chemical information. The Sb sensilla of *Locusta migratoria manilensis* (Megen) (Orthoptera: Locustidae) can sense the species’ aggregation pheromone [30]. Sb sensilla were found in large numbers on the mouthparts of *A. hygrophila*. While the Sb sensillum is generally considered to be an olfactory receptor that responds to odor molecules, it is likely to also function as a taste receptor. For example, Sb sensilla on the maxillary palp of locusts can sense plant aldehydes [31], and Sb sensilla on the fistula of *Papilio xuthus* Linnaeus (Lepidoptera: Papilionidae) can sense sucrose solutions [32]. In addition, Sb sensilla act in mechanical perception, including physical contact, body tension, and ambient temperature and humidity [17]; Sb sensilla on the ovipositor of *Carposina niponensis* Walsingham (Lepidoptera: Carposinidae) sense temperature and humidity and help in the selection of oviposition hosts [33,34].

The St sensillum was one of the most abundant sensilla in *A. hygrophila* adults. It had an apical hole similar to that found in most beetles [35,36,37]. These sensilla can detect various types of sensations, including mechanical, chemical, and heat stimuli. St sensilla on the ovipositor of *Helicoverpa assulta* (Guenée) (Lepidoptera: Noctuidae) are involved in the selection of oviposition hosts [38]; St sensilla on the tarsus of the scale *Quadrastichus erythrinae* Kim (Hymenoptera: Eulophidae) have a similar function [39].

Sch sensilla were the most widely distributed and the longest sensilla on the antennae of *A. hygrophila* and were also the most abundant sensilla on the tarsus. Sch are found on the surface of the epidermis and are generally considered mechanical and chemical sensilla [40]. Studies have found that each Sch sensillum is controlled by one or two bipolar nerve cells and that there are dendrites of sensory neurons in its apical pore. Therefore, it is believed that Sch sensilla also have olfactory [41] and gustatory functions [42]. For example, Sch sensilla on the antennae of *Helicoverpa armigera* (Hübner) (Lepidoptera: Noctuidae) respond to fructose [43]. We infer that these sensilla in *A. hygrophila* may be involved in the selection of host plants, mates, and oviposition sites and would probably be an important target for studying the taxis mechanisms of insects. 

Böhm’s bristles (BB) and sensilla campaniform (Sca) were distributed on the antennae and mouthparts of *A. hygrophila* adults. In beetles, Böhm’s bristles generally occur either singly or in pairs at the base of the antennal scape and in the internode between the pedicel and scape [35,44,45], which was the case for *A. hygrophila*. Böhm’s bristles can sense mechanical stimulation and contribute to regulating the motion and position of the antennae [22,46]. Sca sensilla, in addition to detecting mechanical stimuli [47], are sensitive to carbon dioxide, odors, and changes in temperature or humidity, which has been confirmed in the carabid beetles *Bembidion properans* (Stephens) and *Platynus assimilis* Paykull [48,49].

Sensilla coeloconica (Sco) and sensilla styloconica (Sty) were found on segments 2–9 and 6–9 of the antennal flagellum apex in *A. hygrophila*. It is generally believed that these sensilla have dual functions as both olfactory and temperature or humidity receptors [44,50,51,52]. For example, these two kinds of sensilla on *Bactrocera tryoni* (Froggatt) (Diptera: Tephritidae) [53] are temperature and humidity detectors. In *Drosophila mojavensis* (Patterson and Crow) (Diptera: Drosophilidae), sensilla coeloconica have olfactory functions [54]. Medial Sty sensilla on the maxilla of larvae of *Plutella xylostella* (L.) (Lepidoptera: Plutellidae) [55] and *H. armigera* [56] can play a role in gustation. These receptors may have similar functions in *A. hygrophila*.

Five additional kinds of sensilla were identified in *A. hygrophila* adults: sensilla terminal (S.te), sensilla aperture (Sa), sensilla dome (Dom), sensilla digit-like (Sdi), and a newly discovered one, sensilla petal-shaped (Sps). Sensilla terminal (S.te) are generally considered to be contact chemical sensilla with a gustatory function [57]. Sensilla aperture (Sa) are widely found in beetles [58,59] and have an olfactory function, which has been confirmed in some parasitoids (Hymenoptera) [60]. Sensilla dome (Dom) and sensilla digit-like (Sdi) are generally considered to be mechanical receptors, though probably with the additional function of temperature and humidity receptors [61,62,63,64]. Here we reported a new type of receptor in adults of *A. hygrophila*, named here as sensilla petal-shaped (Sps). These sensilla were located on the terminal segment of the maxillary palps, mainly on the epidermis of the distal segment. These sensilla have a large pore. In general, insect olfaction receptors have a porous cuticle, with structures that allow the entry of odor molecules. For example, the perforated structure at the distal segment of the maxillary palp of *Dianous coeruleotinctus* Puthz (Coleoptera: Staphylinidae) is believed to be a chemoreceptor [65]. Similarly, the pores in the epipharyngeal sensilla of the larva of *M. sexta* are an important taste receptor used to guide host feeding [66,67,68]. The newly discovered receptor, sensilla petal-shaped (Sps), has similar structural features. Therefore, based on the morphological characteristics of this new sensilla, we inferred that it may have olfactory and gustatory functions, which needs further study.

### 4.2. Sexual Dimorphism of A. hygrophila Adults

Similar to other beetles [69], the adults of *A. hygrophila* show obvious sexual dimorphism in external morphology. Females have a larger body size with a rounded ventral genital segment, while the equivalent male segment is slightly dented in shape. The sensilla attached to the body surfaces of males and females are also different, both on the head and the tarsi; differences include the subtypes present and the size of the sensilla. The morphology of the tarsal attachment pads of males vs. females and the size of their tarsal sensilla are also different between the sexes. On the head, there were more types of antennal sensilla in females than in males. In *A. hygrophila*, types Sty.3 and Sco.3 were found only in male antennae, while types Sb4, Sco.1–2, Sty.2, BB.2, and Sca.1 were found only on females’ antennae. 

In addition, the expression of some genes plays a role in the sexual dimorphism of sensilla. Previous studies have shown that some odor receptor genes of *A. hygrophila* tend to be highly expressed in the antennae of females [20]. This gender bias is also found in other insects. The high expression in females of the olfactory receptor genes of the antennae of *Bombyx mori* L. (Lepidoptera: Bombycidae) and *Spodoptera exigua* (Hübner) (Lepidoptera: Noctuidae) may help the females find suitable oviposition sites on the host plant [70,71]. However, males of both *H. armigera* and *H. assulta* have high expression of olfactory receptor genes in their antennae, which may be related to mating behavior [72]. Therefore, we speculated that the genes specifically expressed in the sensilla of male or female *A. hygrophila* may also have gender-specific functions. In addition, we observed sexual dimorphism in tarsal attachment pads of *A. hygrophila*, principally in the shape of the pads on the fore tarsus, which is spoon-shaped on males versus a curved hook shape for females. These microscopic indicators have value for sex identification.

### 4.3. Sensilla and Monophagy of A. hygrophila Adults

In our study, a total of 12 types of sensilla were identified in *A. hygrophila* adults, including many subtypes, such as Sb with 12 subtypes and St with 8 subtypes. This diversity presents a more complicated situation than was previously known. It is generally believed that the sensory complexity of monophagous insects is lower than that of polyophagous insects [73]. Monophagous and oligophagous insects often use plant secondary substances as “signature stimulus” for host selection. For example, some species of moths/butterflies, flies, and beetles use the unique secondary substances of their Cruciferae or Umbelliferae hosts as their signal substances for host identification [74]. Therefore, the response of monophagous insects to feeding stimuli is more specific, faster, more accurate, and more efficient than that of polyphagous insects [75]. Some monophagous insects also have specialized sensilla that can sense the specific secondary substances indicative of their hosts. For example, *M. sexta* initiates feeding only when the taste receptor cells receive specific and adequate chemical sensory information [76,77]. We also found that the host selection behavior of *A. hygrophila* was influenced by the volatiles of specific host plants [17]; however, the specific function of the receptors still needs to be further explored. In addition, compared with other oligophagous or even monophagous beetles such as *Luperomorpha sururalis* Chen (Coleoptera: Chrysomelidae) [78] and *Scythropus yasumatsui* Koneet Merimoto (Coleoptera: Curculionidae) [79], the number of types of *A. hygrophila* sensilla is greater. It is generally believed that the insect surface receptors play a key role in the chemical recognition of host plants, and their types are related to the habits of the host plant, its habitat, and the plant’s spatial distribution. For monophagous insects, only a few chemicals need to be detected for host recognition, and the types of sensilla may therefore be relatively low [7]. However, the monophagous species *A. hygrophila* has more, not fewer, sensilla types. It is speculated that there may be dynamic changes in the host range during long-term co-evolution and species differentiation with host plants, which supports Janz’s “oscillation hypothesis” of host range and species evolution [80]. Therefore, comparing the specific differences between species, such as the type and number of sensilla, and even the gene expression within sensilla may not clearly determine the insect’s degree of specialization, a general finding that has been confirmed in the fruit fly *D. mojavensis* [55].

## 5. Conclusions

This study provides the first detailed fine morphological characterization of *A. hygrophila* and the fine structure, location, and distribution of different sensilla types in both males and females. Twelve types and forty-six subtypes of sensilla were observed, some of which showed obvious sexual dimorphism. In particular, a new type of sensor on the distal segment of the maxillary palps was reported for the first time, named sensilla petal-shaped (Sps), whose function may be related to host recognition. This hypothesis needs to be verified by transmission electron microscopy and single sensillum recording. This work provides a foundation for subsequent studies in chemical ecology, electrophysiology, and behavior and may contribute to future studies of host recognition mechanisms in monophagous insects.

## Figures and Tables

**Figure 1 insects-14-00501-f001:**
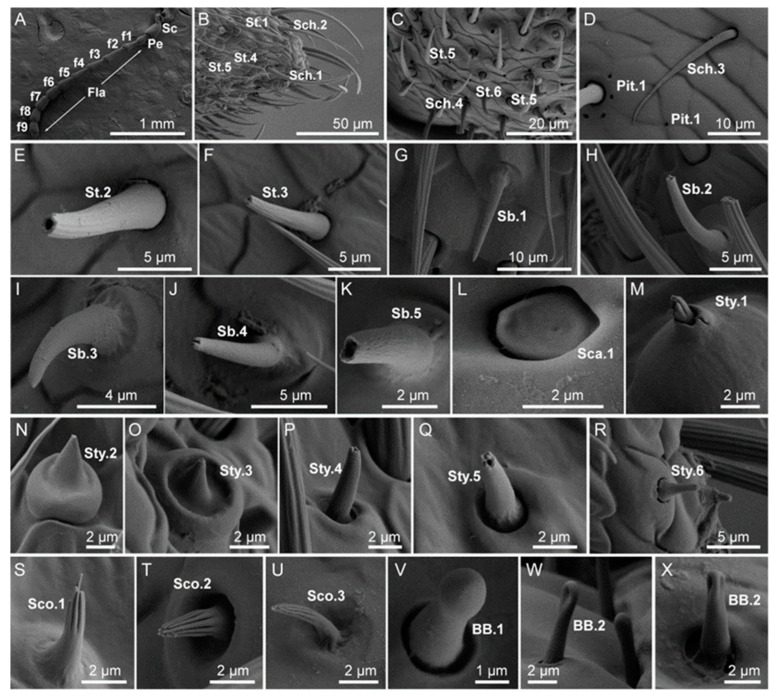
(**A**) Morphology of *Agasicles hygrophila* adult antennae; Sc, scape; Pe, pedicel; Fla, flagellum. SEM structure of antennal sensilla of *A. hygrophila*. (**B**) Sensilla chaetica (Sch.1, Sch.2), sensilla trichodea (St.1, St.4, St.5); (**C**) sensilla chaetica (Sch.4), sensilla trichodea (St.5, St.6); (**D**) sensilla chaetica (Sch.3); (**E**) sensilla trichodea (St.2); (**F**) sensilla trichodea (St.3); (**G**) sensilla basiconca (Sb.1); (**H**) sensilla basiconca (Sb.2); (**I**) sensilla basiconca (Sb.3); (**J**) sensilla basiconca (Sb.4); (**K**) sensilla basiconca (Sb.5); (**L**) sensilla campaniform (Sca.1); (**M**) sensilla styloconica (Sty.1); (**N**) sensilla styloconica (Sty.2); (**O**) sensilla styloconica (Sty.3); (**P**) sensilla styloconica (Sty.4); (**Q**) sensilla styloconica (Sty.5); (**R**) sensilla styloconica (Sty.6); (**S**) sensilla coeloconica (Sco.1); (**T**) sensilla coeloconica (Sco.2); (**U**) sensilla coeloconica (Sco.3); (**V**) Böhm bristles (BB.1); (**W**) Böhm bristles (BB.2); (**X**) Böhm bristles (BB.2).

**Figure 2 insects-14-00501-f002:**
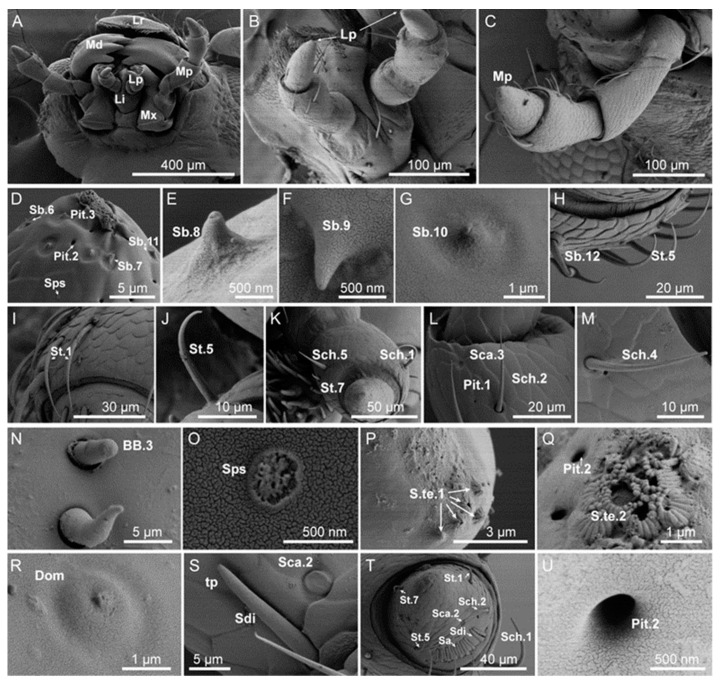
(**A**) Morphology of *Agasicles hygrophila* adult mouthparts: Lr, labrum; Md, mandible; Mx, maxilla; Li, labium; Mp, maxillary palp; Lp, labial palp; (**B**) Lp, labial palp; (**C**) Mp, maxillary palp. SEM structure of maxillary and labial palps sensilla of *A. hygrophila*: (**D**) sensilla basiconca (Sb.6, Sb.7, Sb.11), sensilla terminal (S.te.2), sensilla petal-shaped (Sps), Pit.2, Pit.3; (**E**) sensilla basiconca (Sb.8); (**F**) sensilla basiconca (Sb.9); (**G**) sensilla basiconca (Sb.10); (**H**) sensilla trichodea (St.5), sensilla basiconca (Sb.12); (**I**) sensilla trichodea (St.1), Pit.1; (**J**) sensilla trichodea (St.5); (**K**) sensilla chaetica (Sch.1, Sch.5), sensilla trichodea (St.7); (**L**) sensilla chaetica (Sch.2), sensilla campaniform (Sca.3), Pit.1; (**M**) sensilla chaetica (Sch.4); (**N**) Böhm bristles (BB.3); (**O**) sensilla petal-shaped (Sps); (**P**) sensilla terminal (S.te.1), Pit.3; (**Q**) sensilla terminal (S.te.2), Pit.2 (ST); (**R**) sensilla dome (Dom); (**S**) sensilla digit-like (Sdi), tp highlights the top pit at the end of the Sdi, sensilla campaniform (Sca.2); (**T**) terminal segment, sensilla chaetica (Sch.1, Sch.2), sensilla trichodea (St.1, St.5, St.7), sensilla campaniform (Sca.2), sensilla aperture (Sa), sensilla digit-like (Sdi); (**U**) Pit.2, the pit around the end of the distal segment of the labial palps.

**Figure 3 insects-14-00501-f003:**
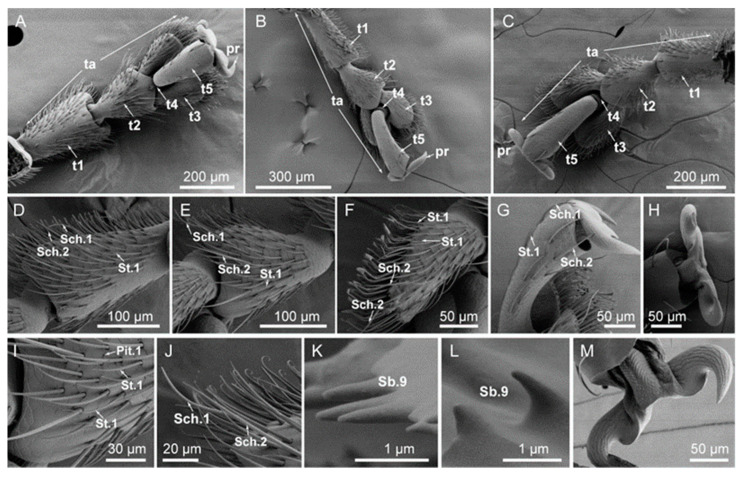
(**A**) Morphology of the fore tarsus of an *Agasicles hygrophila* adult: t1–t5, tarsites; ta, tarsus; pr, pretarsus (ungues). (**B**) Morphology of the mid-tarsus of an *A. hygrophila* adult: t1–t5, tarsites; ta, tarsus; pr, pretarsus (ungues). (**C**) Morphology of the hind tarsus of an *A hygrophila* adult: t1–t5, tarsites; ta, tarsus; pr, pretarsus (ungues). (**D**) Morphology of sensilla of the first tarsal segment of an adult *A. hygrophila*: sensilla chaetica (Sch.1, Sch.2), sensilla trichodea (St.1). (**E**) Morphology of the second tarsal segment of an adult *A. hygrophila*: sensilla chaetica (Sch.1, Sch.2), sensilla trichodea (St.1). (**F**) Morphology of the third tarsal segment of an adult of *A. hygrophila*: sensilla chaetica (Sch.2), sensilla trichodea (St.1). (**G**) Morphology of the fifth tarsal segment of an adult *A. hygrophila*: sensilla chaetica (Sch.1, Sch.2), sensilla trichodea (St.1). (**H**) Morphology of the pretarsus of the forefoot and the midfoot. (**I**) Sensilla trichodea (St.1), Pit.1. (**J**) Sensilla chaetica (Sch.1, Sch.2). (**K**) General view of sensilla basiconca (Sb.9) of the fifth tarsal segment. (**L**) General view of the sensilla basiconca (Sb.9) of the basal portion of the pretarsus. (**M**) Morphology of the pretarsus of the hindfoot.

**Figure 4 insects-14-00501-f004:**
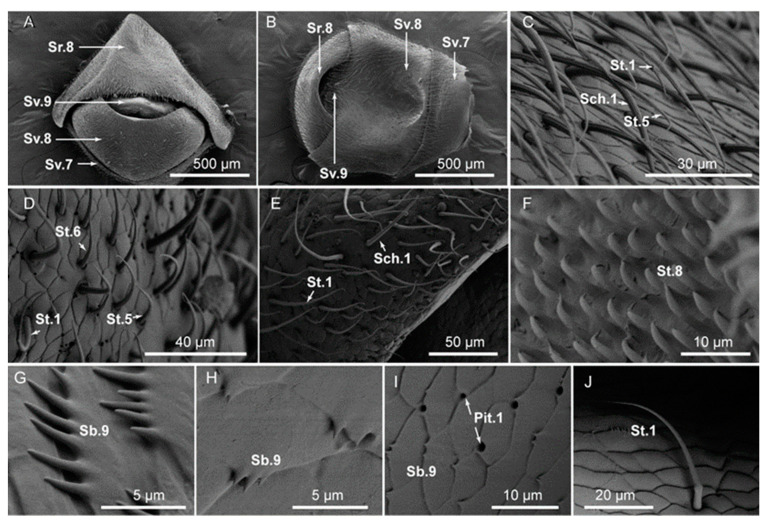
(**A**) Morphology of the female external genital segments in adults of *Agasicles hygrophila*: sv.7 on venters of genital segment 7; sv.8 on venters of genital segment 8; sv.9 on the venter of genital segment 9; sr.8 on the posterior of abdominal segment 8. (**B**) Morphology of male external genital segments in *A. hygrophila* adults. (**C**) Sensilla trichodea (St.1, St.5), sensilla chaetica (Sch.1). (**D**) Sensilla trichodea (St.1, St.5, St.6). (**E**) Sensilla trichodea (St.1), sensilla chaetica (Sch.1). (**F**) Sensilla trichodea (St.8). (**G**–**I**) Sensilla basiconca (Sb.9). (**J**) Sensilla trichodea (St.1).

**Table 1 insects-14-00501-t001:** Main sensilla differences in antennae between male and female *Agasicles hygrophila* adults.

Type of Sensillum	Gender	Number	Length (μm)	Basal Width (μm)	Subtype	Location
Sensilla chaeticum	Female	803.36 *	90.49–39.45 *	5.65–2.37 *	4	S,P,F1-F9
	Male	761.33 *	80.11–17.70 *	4.15–1.62 *	4	S,P,F1-F9
Sensilla trichodeum	Female	913.07 *	46.33–8.96 *	3.76–2.04 *	6	P,F1-F9
	Male	1221.67 *	37.20–12.26 *	3.07–2.07 *	6	P,F1-F9
Sensilla basiconicum	Female	859.67 *	14.2–6.6 *	5.97–1.56 *	5	F2-F9
	Male	776.33 *	11.33–4.25 *	2.86–1.69	4 (no Sb.4)	F1-F9
Sensilla coeloconica	Female	103.33	4.90–3.29 *	1.84–1.39 *	4 (no Sb.3)	F2-F9
	Male	100.33	2.97–2.93 *	1.54–1.50 *	1 (no Sco.1, Sco.2)	F4-F8
Sensilla styloconica	Female	255.35 *	5.44–1.18 *	2.78–0.97 *	5 (no Sty.3)	F2-F9
	Male	181.00 *	4.73–1.49	1.97–0.95 *	5 (no Sty.2)	F2-F9
Sensilla campaniform	Female	4.00 *	—	2.26–1.45 *	1	F9
	Male	—	—	—	—	—

Note: * means there are significant differences on the *p* = 0.05 level between females and males using Student’s *t* tests.

**Table 2 insects-14-00501-t002:** Main sensilla differences in labial palps between adult male and female *Agasicles hygrophila*.

Type of Sensillum	Gender	Number	Length (μm)	Basal Width (μm)
Sensilla chaeticum	Female	3	45.28–36.96 *	5.63–1.90 *
	Male	4	53.14–27.80 *	3.55–2.45 *
Sensilla trichodeum	Female	7	16.63–12.48 *	2.01–2.00
	Male	3	23.35–10.40 *	2.20–2.06
Sensilla basiconicum	Female	9	0.81–0.37	0.68–0.55 *
	Male	12	0.94–0.37	0.77–0.34 *
Sensilla terminal	Female	10	0.39–0.43	0.69–0.65 *
	Male	12	0.43–0.45	0.61–0.59 *
Sensilla campaniform	Female	2	—	3.06–2.92 *
	Male	3	—	3.22–3.13 *
Sensilla dome	Female	—	—	—
	Male	2	0.28–0.27	0.31–0.30

Note: * means there are significant differences on the *p* = 0.05 level between females and males using Student’s *t* tests.

**Table 3 insects-14-00501-t003:** Main sensilla differences in maxillary palps between adult male and female *Agasicles hygrophila*.

Type of Sensillum	Gender	Number	Length (μm)	Basal Width (μm)	Subtype
Sensilla chaeticum	Female	7	66.20–17.04 *	5.60–1.56 *	3
	Male	8	101.54–20.60 *	3.93–1.12 *	3
Sensilla trichodeum	Female	42	52.05–9.14 *	4.13–1.27 *	3
	Male	34	23.59–6.61 *	3.23–1.39 *	3
Sensilla basiconicum	Female	59.23	0.86–0.68	1.16–0.41 *	6 (no Sb.8)
	Male	63.47	0.97–0.73	0.90–0.59 *	6 (no Sb.7)
Sensilla terminal	Female	14	—	0.51–0.38 *	1
	Male	18	—	0.78–0.73 *	1
Sensilla petal-shaped	Female	3	—	0.37–0.36 *	1
	Male	4	—	1.05–1.03 *	1

Note: * means there are significant differences on the *p* = 0.05 level between females and males using Student’s *t* tests.

## Data Availability

All data generated or analyzed during this study are included in this published article.

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
