# Peer review of "Morphological Characterstics of the Sensilla in a Monophagous Insect: *Agasicles hygrophila* Selman and Vogt (Coleoptera: Chrysomelidae, Halticinae)"

_insects, 2023, doi:10.3390/insects14060501_

Round 1

Reviewer 1 Report

In the introduction the manuscript background is correctly given, and no further information are required

In the M&M section the indication of the collection locality of the material should be given. The material was bought from the quoted firm, but any further information should be added.

at row 127 there is a full stop (Crook.), please check everywhere in the manuscript for any other misspelling.

The description of the sensilla is quite detailed, the images are of good quality. No changes are required for the result section.

Discussion and Conclusions meet the results.

Author Response

Comment 1: In the introduction the manuscript background is correctly given, and no further information are required.

Response: Thank you for your positive comments on our manuscript.

Comment 2: In the M&M section the indication of the collection locality of the material should be given. The material was bought from the quoted firm, but any further information should be added.

Response: Thanks for your suggestion on improving the accessibility of our manuscript. We have added the collection locality of the material in Line 118-120 of the updated manuscript.

Comment 3: at row 127 there is a full stop (Crook.), please check everywhere in the manuscript for any other misspelling.

Response: Thank you for your reminder, we were sorry for our careless mistakes. According to your suggestion, we have removed the full stop here and checked the spelling throughout.

Comment 4: The description of the sensilla is quite detailed, the images are of good quality. No changes are required for the result section. Discussion and Conclusions meet the results.

Response: Thank you for your positive comments and valuable suggestions to improve the quality of our manuscript.

Reviewer 2 Report

Dear Authors,

here you can find my comments on your paper entitled: “Morphology and Ultrastructure of the Sensilla of a Monophagous Insect: Agasicles hygrophila Selman & Vogt (Coleoptera: Chrysomelidae, Halticinae)”.

I appreciated your article that is interesting with a detailed description of numerous sensilla. It is supported by clear images and statistic information enriches your work. I recommend it for publication in “Insect”.

Nevertheless, I found inappropriate the use of words like “ultrastructure” or “fine morphology” or “ultramorphology” etc, since you referred to the external structure of sensilla without an investigation on nervous system of each sensillum. I believe you should avoid these words along all the MS, limiting to “morphology”, especially in the title and keywords.

If possible, I would like you to provide a more detailed image of sensillum petal shape in order to extend the resonance of this interesting finding.

I provided also some minor suggestions to improve your MS.

SIMPLE SUMMARY

Line 14: “of this species in both in male and female adults” delete “in” between both and male.

Line 14-15: change “showed” with “displayed” or something similar to avoid repetitions.

Line 16: delete comma between “and” and “named” and change "sensilla" in "Sensilla", with a capital letter.

Line 17: delete comma between “sensor” and “may be”.

Line 19: change “lay” in “lays”.

ABSTRACT

Line 23: evaluate to delete “the” before “A. hygrophila”.

Line 24 and so on along the abstract: I suppose that abbreviation names (e.g. (SEM), (Sch), (St), (Sb)…) are not necessary in this section. Please, consider removing them.

Line 26: “included” and “including” are repeated in the same sentence. Please, change one.

Line 31: modify “was named the sensilla” with “was named as sensilla”.

Line 32: after removing acronyms in this section (see comment before), change "Sch", "St", and "Sb" with their extended forms.

Line 33 and 35: “coeloclnica” is not well-written.

Line 35: change “in addition” with “on the contrary”.

Line 36: Delete “and the” in the sentence: “males. And the numbers”.

Line 38: change “investigations of beetles” with “investigations on beetles”.

INTRODUCTION

Line 48: delete “worldwide” since it is repeated also in lines 46 and 49.

Line 54: “the related species, Alternanthera sessilis (L.) DC, [5]”. Please delete commas before “Alternanthera” and after “DC,”. Additionally, change “this use” in “the use”.

Lines 55-56: “risk of reduction” and “or of potential harm”. Change both “of” in “on”.

Lines 61-66: I think that these statements require some references. Please add some citations.

Line 63: modify “Sensilla of many kinds” in “Many kinds of sensilla”.

Line 65: substitute “many types” in “several types” to avoid the repetition of “many”.

Line 89: delete “for A. hygrophila adults” and add “of A. hygrophila adults” after “females and males”.

Line 94: change “investigation of the mechanism” in “investigation on the mechanism”.

Line 95: “understanding of the relationship” change in “understanding on the relationship”.

MATERIALS AND METHODS

Line 99: consider to change the subchapter title in “insect collection”.

Line 100: add “specimens”, or something similar, after “Agasicles hygrophila”. Consequently, change “was obtained” in “were obtained”.

Lines 103-104: do not use the contract form for Agasicles hygrophila and A. philoxeroides at the beginning of the sentences, please extend genus names.

Line 104: move “was maintained” at the end of the sentence and change it in “were maintained”.

Line 107: modify “light: dark = 14:10 h” in “a 10 h dark–14 h light photoperiod”.

Line 109: consider changing the subchapter title in “SEM procedures”.

Line 110: Are the insects randomly selected from the colony? In this case, add “randomly” in the sentence. Besides, please provide information on the total number of examined specimens, and how many females and males.

Lines 126-129: please check if these citations are reported following journal rules. I believe Schneider (1964) [20] should be written as “Schneider [20]”. Besides, add a space between “Crook” and “et al.”.

Lines 129-130: add “to” after regards.

RESULTS

Line 148: consider changing the sentence in “the total length of the antenna”.

Line 152: correct the space between “subtypes” and “St”.

Line 156: please refer to the subtypes.

Table 1 and 2: I’m wondering why some sensillum names are in singular form, while others in plural.

Line 179: please add a dot between “(Table 2)” and “The maxillary palps”.

Line 215: change “SCH” in “Sch”.

Line 244: change “ST” in “St”.

Line 279: change “SB” in “Sb”.

Lines 310, 317, 423: delete “were”.

Lines 412, 454: move the dot at the end of the sentence and delete it after “antenna” (412) and “base” (454).

Line 477 subchapter 3.3: I would prefer you to use all verbs in the same form (present or past).

Line 555: I think that Aedes aegypyi is not an appropriate example since it is a hematophagous insect with a host location mechanism totally different compared to phytophagous insects. I suggest you to remove this species.

Line 583: I believe a word is missing in the sentence “the female has larger”…?

Line 589: add a dot between “males” and “Types”.

Lines 643-644: acronyms TEM and SSR are not necessary, please delete.

FIGURES AND TABLES

I suggest you to write in the extended form the name species in the captions.

The MS is well-written and just some corrections are needed.

Author Response

Comment 1: I appreciated your article that is interesting with a detailed description of numerous sensilla. It is supported by clear images and statistic information enriches your work. I recommend it for publication in “Insect”. Nevertheless, I found inappropriate the use of words like “ultrastructure” or “fine morphology” or “ultramorphology” etc, since you referred to the external structure of sensilla without an investigation on nervous system of each sensillum. I believe you should avoid these words along all the MS, limiting to “morphology”, especially in the title and keywords.

Response: Thank you for your positive comments and valuable suggestions to improve the quality of our manuscript. We have addressed your suggestions after discussion and changed the title to "Morphological characterstics of the Sensilla in a Monophagous Insect: Agasicles hygrophila Selman & Vogt (Coleoptera: Chrysomelidae, Halticinae)"(Line 2). And then we replaced "ultrastructure" in keywords by "morphology" (Line 42). In other parts of the article, we replaced "ultrastructure" in Line 23 with "morphology", modified "fine morphology" in Line 24 with "morphological characteristics", replaced "ultra-microstructure" in Line 89 with "morphological", and deleted "at the ultra-fine level" in Line 96 and "ultra" in Line 97.

Comment 2: If possible, I would like you to provide a more detailed image of sensillum petal shape in order to extend the resonance of this interesting finding.

Response: Thank you for this suggestion. This image in our manuscript is the most detailed image. We are sorry that we cannot provide a more detailed image in a short time because scanning electron microscopy requires an appointment.

Comment 3: I provided also some minor suggestions to improve your MS.

Response: Thank you very much for your questions. We have addressed those questions and suggestions in the Responses to Comment 4-52, and we have numbered them for your convenience.

SIMPLE SUMMARY:

Comment 4: Line 14: “of this species in both in male and female adults” delete “in” between both and male.

Response: Thank you for pointing this out. It has been modified.

Comment 5: Line 14-15: change “showed” with “displayed” or something similar to avoid repetitions.

Response: Thanks for your suggestion. It has been modified in Line 15.

Comment 6: Line 16: delete comma between “and” and “named” and change "sensilla" in "Sensilla", with a capital letter.

Response: They have been modified.

Comment 7: Line 17: delete comma between “sensor” and “may be”.

Response: It has been modified.

Comment 8: Line 19: change “lay” in “lays”.

Response: It has been modified.

ABSTRACT:

Comment 9: Line 23: evaluate to delete “the” before “A. hygrophila”.

Response: It has been modified.

Comment 10: Line 24 and so on along the abstract: I suppose that abbreviation names (e.g. (SEM), (Sch), (St), (Sb)…) are not necessary in this section. Please, consider removing them.

Response: Thank you for the detailed review. We have removed all abbreviations that appear in the abstract.

Comment 11: Line 26: “included” and “including” are repeated in the same sentence. Please, change one.

Response: Thanks for your suggestion. we have changed "include" in Line 26 to "contain".

Comment 12: Line 31: modify “was named the sensilla” with “was named as sensilla”.

Response: It has been modified.

Comment 13: Line 32: after removing acronyms in this section (see comment before), change "Sch", "St", and "Sb" with their extended forms.

Response: They have been modified in Line 32.

Comment 14: Line 33 and 35: “coeloclnica” is not well-written.

Response: Thank you for pointing this out. We have corrected "coeloclnica" to "coeloconica" in Line 34 and 35.

Comment 15: Line 35: change “in addition” with “on the contrary”.

Response: Thanks for your suggestion, it has been modified.

Comment 16: Line 36: Delete “and the” in the sentence: “males. And the numbers”.

Response: It has been modified.

Comment 17: Line 38: change “investigations of beetles” with “investigations on beetles”.

Response: It has been modified.

INTRODUCTION:

Comment 18: Line 48: delete “worldwide” since it is repeated also in lines 46 and 49.

Response: It has been modified.

Comment 19: Line 54: “the related species, Alternanthera sessilis (L.) DC, [5]”. Please delete commas before “Alternanthera” and after “DC,”. Additionally, change “this use” in “the use”.

Response: They have been modified.

Comment 20: Lines 55-56: “risk of reduction” and “or of potential harm”. Change both “of” in “on”.

Response: They have been modified.

Comment 21: Lines 61-66: I think that these statements require some references. Please add some citations.

Response: Thank you for the detailed review. We have added citations [8] and [9] in these statements to better support our point, the references information is showed as below.

[8] Miller, J.R.; Stricker, K.L. Finding and accepting host plants. Bell, W.J; Cardé; R.T. The Chemical Ecology of Insects. London: Campman and Hall, 1984. Chapter 6, 125-157.

[9] Zacharuk, R.Y. Ultrastructure and function of insect chemosensilla. Annu. Rev. Entomol. 1980, 25, 2747.

Comment 22: Line 63: modify “Sensilla of many kinds” in “Many kinds of sensilla”.

Response: It has been modified.

Comment 23: Line 65: substitute “many types” in “several types” to avoid the repetition of “many”.

Response: It has been modified.

Comment 24: Line 89: delete “for A. hygrophila adults” and add “of A. hygrophila adults” after “females and males”.

Response: They have been modified.

Comment 25: Line 94: change “investigation of the mechanism” in “investigation on the mechanism”.

Response: It has been modified.

Comment 26: Line 95: “understanding of the relationship” change in “understanding on the relationship”.

Response: It has been modified.

MATERIALS AND METHODS:

Comment 27: Line 99: consider to change the subchapter title in “insect collection”.

Response: It has been changed to "Insect Collection".

Comment 28: Line 100: add “specimens”, or something similar, after “Agasicles hygrophila”. Consequently, change “was obtained” in “were obtained”.

Response: They have been modified.

Comment 29: Lines 103-104: do not use the contract form for Agasicles hygrophila and A. philoxeroides at the beginning of the sentences, please extend genus names.

Response: Thank you for pointing this out. They have been modified in Line 106 and 108.

Comment 30: Line 104: move “was maintained” at the end of the sentence and change it in “were maintained”.

Response: It has been modified.

Comment 31: Line 107: modify “light: dark = 14:10 h” in “a 10 h dark–14 h light photoperiod”.

Response: It has been modified.

Comment 32: Line 109: consider changing the subchapter title in “SEM procedures”.

Response: It has been modified.

Comment 33: Line 110: Are the insects randomly selected from the colony? In this case, add “randomly” in the sentence. Besides, please provide information on the total number of examined specimens, and how many females and males.

Response: Thank you for your positive comments and valuable suggestions to improve the quality of our manuscript. We have added the details "Newly emerged A. hygrophila adults (six males and six females) were selected randomly from the laboratory colony" in Line 114.

Comment 34: Lines 126-129: please check if these citations are reported following journal rules. I believe Schneider (1964) [20] should be written as “Schneider [20]”. Besides, add a space between “Crook” and “et al.”.

Response: They have been modified.

Comment 35: Lines 129-130: add “to” after regards.

Response: It has been modified.

RESULTS:

Comment 36: Line 148: consider changing the sentence in “the total length of the antenna”.

Response: Thanks for your suggestion. We have replaced "the length of the total antenna of the female adult" with "the length of total antenna of female A. hygrophila" in the manuscript.

Comment 37: Line 152: correct the space between “subtypes” and “St”.

Response: It has been modified.

Comment 38: Line 156: please refer to the subtypes.

Response: Thank you for the detailed review. We have changed the sentence order in Lines 162-166, and added the numbers of subtypes found in different genders to be easy to read.

Comment 39: Table 1 and 2: I’m wondering why some sensillum names are in singular form, while others in plural.

Response: Thank you for your reminding. We checked the results and believed that the plural form should be used. We have changed "sensillum" to "sensilla".

Comment 40: Line 179: please add a dot between “(Table 2)” and “The maxillary palps”.

Response: It has been modified.

Comment 41: Line 215: change “SCH” in “Sch”.

Response: It has been modified.

Comment 42: Line 244: change “ST” in “St”.

Response: It has been modified.

Comment 43: Line 279: change “SB” in “Sb”.

Response: It has been modified.

Comment 44: Lines 310, 317, 423: delete “were”.

Response: They have been modified.

Comment 45: Lines 412, 454: move the dot at the end of the sentence and delete it after “antenna” (412) and “base” (454).

Response: It has been modified.

Comment 46: Line 477 subchapter 3.3: I would prefer you to use all verbs in the same form (present or past).

Response: Thank you for pointing this out. We have unified all verb tenses in subchapter 3.3 into the simple past tense.

Comment 47: Line 555: I think that Aedes aegypyi is not an appropriate example since it is a hematophagous insect with a host location mechanism totally different compared to phytophagous insects. I suggest you to remove this species.

Response: Thank you for the detailed review. It has been removed.

Comment 48: Line 583: I believe a word is missing in the sentence “the female has larger”…?

Response: Thanks for your suggestion on improving the accessibility of our manuscript. We have added "body size" after "the female has larger".

Comment 49: Line 589: add a dot between “males” and “Types”.

Response: It has been modified.

Comment 50: Lines 643-644: acronyms TEM and SSR are not necessary, please delete.

Response: It has been modified.

FIGURES AND TABLES

Comment 51: I suggest you to write in the extended form the name species in the captions.

Response: Thank you again for your positive comments and valuable suggestions to improve the quality of our manuscript. We have carefully reviewed the titles of all the charts and tables in the manuscript and changed the species names to extensions in Lines 168, 179, 204, 217, 220, 474, 505, 679, 682, 684, and Line 687.

Comments on the Quality of English Language

Comment 52: The MS is well-written and just some corrections are needed.

Response: Thank you for your positive comments on our manuscript. We have revised every problem pointed out according to your comments. We believe that the quality of the manuscript has also been improved, and hope that it can meet your requirements.